# A Multi-Scale Anti-Multipath Algorithm for GNSS-RTK Monitoring Application

**DOI:** 10.3390/s23208396

**Published:** 2023-10-11

**Authors:** Shouhua Wang, Shuaihu Wang, Xiyan Sun

**Affiliations:** 1College of Information and Communication, Guilin University of Electronic Technology, Guilin 541004, China; wangxhca@163.com (S.W.); sunxiyan1@163.com (X.S.); 2Key Laboratory of Cognitive Radio and Information Processing, School of Information and Communication, Guilin University of Electronic Technology, Ministry of Education, Guilin 541004, China; 3National Engineering Research Center for Satellite Navigation, Positioning and Position Service, Guilin 541004, China

**Keywords:** GNSS-RTK, multipath error, ICEEMDAN, wavelet packet denoising, adaptive thresholding denoising

## Abstract

During short baseline measurements in the Real-Time Kinematic Global Navigation Satellite System (GNSS-RTK), multipath error has a significant impact on the quality of observed data. Aiming at the characteristics of multipath error in GNSS-RTK measurements, a novel method that combines improved complete ensemble empirical mode decomposition with adaptive noise (ICEEMDAN) and adaptive wavelet packet threshold denoising (AWPTD) is proposed to reduce the effects of multipath error in GNSS-RTK measurements through modal function decomposition, effective coefficient sieving, and adaptive thresholding denoising. It first utilizes the ICEEMDAN algorithm to decompose the observed data into a series of intrinsic mode functions (IMFs). Then, a novel IMF selection method is designed based on information entropy to accurately locate the IMFs containing multipath error information. Finally, an optimized adaptive denoising method is applied to the selected IMFs to preserve the original signal characteristics to the maximum possible extent and improve the accuracy of the multipath error correction model. This study shows that the ICEEMDAN-AWPTD algorithm provides a multipath error correction model with higher accuracy compared to singular filtering algorithms based on the results of simulation data and GNSS-RTK data. After the multipath correction, the accuracy of the E, N, and U coordinates increased by 49.2%, 65.1%, and 56.6%, respectively.

## 1. Introduction

In Global Navigation Satellite System (GNSS) measurements, multipath error occurs due to the presence of reflective sources around the monitored target. This causes the GNSS receiver antenna to simultaneously receive both the direct signal from the satellite and the reflected or diffracted signals, and this leads to deviations in the observed values [1]. Multipath error in the GNSS is generally a result of a lack of spatial correlation and primarily depends on the geometric relationship between the satellite, receiver antenna, and reflecting source. Under certain conditions, the maximum value of multipath error can reach 1/4 of the carrier wavelength. As an example, for the GPS carrier phase, the error of multipath effects can reach 4.8 cm for the L1 carrier and 6.1 cm for the L2 carrier. It is difficult to eliminate through differential techniques and has become the main error source of high-precision positioning [2].

In recent years, the main methods used to mitigate multipath error in GNSS data have been the selection of monitoring station locations, receiver hardware improvements, and software post-processing [3]. Among these methods, software post-processing has been studied extensively by many researchers. Not only does it save costs, but it also greatly weakens the multipath errors. Swathi et al. [4] proposed a multipath correction method based on adaptive filtering, which avoids the problem of losing the effective signal when denoising is applied to the observed data. However, its overall efficiency is relatively low and its performance is unstable. Lau et al. [5] employed the wavelet packet thresholding denoising (WPTD) algorithm to extract a multipath error and then corrected the multipath error in accordance with the repetition period of the satellite orbit. Nevertheless, the hard threshold function used in this algorithm is prone to oscillation of the signal after noise reduction. Yu et al. [6] proposed an improved method by integrating complementary ensemble empirical mode decomposition with adaptive noise (CEEMDAN) and the effective coefficient sieving algorithm. They verified that the proposed algorithm can effectively separate multipath errors and high-frequency noise in GNSS data by using the Tianjin Haihe Bridge as a monitoring target. If the environment around the monitoring station is almost constant over a short period of time, the multipath errors received by the station’s receiver should also be periodic, as the satellite orbit is periodic. On the basis of this property, the multipath errors can be processed by means of the sidereal filter (SF) [7]. Guo et al. [8] proposed a combined method of ensemble empirical mode decomposition (EEMD) and wavelet transform (WT) to mitigate the multipath error in GNSS data and to compensate for the shortcomings of a single EEMD and wavelet transform algorithm. By processing and analyzing the measurement data provided by the GNSS Research Centre at Curtin University, the combined filtering algorithm proved to be more effective in extracting and correcting multipath errors in GNSS data. The current combined algorithm is of great research importance. It retains the advantages of the individual algorithms and compensates to some extent for the disadvantages of using a single algorithm.

Due to the characteristics of multipath errors in GNSS-RTK monitoring, a novel method that combines ICEEMDAN and AWPTD (ICEEMDAN-AWPTD) is proposed to mitigate the multipath errors in GNSS-RTK positioning. First, this paper utilizes the ICEEMDAN algorithm to accurately decompose the coordinate residuals into a series of IMFs and subdivides the IMFs into noise terms, mixed terms, and useful terms according to the proposed partitioning criterion. Second, it employs the AWPTD algorithm to perform noise rejection for the mixed terms and then reconstructs the denoised results with the useful terms to obtain a multipath error correction model. Finally, coordinate data for the following day are corrected using the extracted multipath error. It is demonstrated that the extracted multipath error model is highly accurate and effectively mitigates the multipath errors in GNSS-RTK data.

## 2. Materials and Methods

### 2.1. GNSS-RTK Solving Method and Application

GNSS-RTK technology can largely eliminate satellite ephemeris error, tropospheric and ionospheric error, satellite and receiver clock bias, and other spatially dependent errors in short baseline positioning. It has unique advantages such as high measurement accuracy, high continuity, and the ability to monitor targets in real time. For this, it is widely used to monitor the deformation of large structures such as bridges, high-rise buildings, and dams [9]. To illustrate the working mode of GNSS-RTK in practical applications, Figure 1 shows a schematic diagram of the GNSS-RTK technology used in bridge deformation monitoring.

GNSS-RTK adopts the carrier-phase double-difference dynamic positioning mode and provides high-precision three-dimensional coordinates for monitoring points. Assuming that the co-viewing satellites of the monitoring station (r) and reference station (b) are i and j, and using satellite i as the reference satellite, the carrier-phase double-differential observation model is as follows:(1)$$\lambda \varphi_{br}^{\left(ij\right)}=\rho_{br}^{\left(ij\right)}+\lambda N_{br}^{\left(ij\right)}+M_{\varphi ,br}^{\left(ij\right)}+\varepsilon_{\varphi ,br}^{\left(ij\right)}$$
(2)$$\left\{\begin{array}{l} \rho_{br}^{\left(ij\right)}=\rho_{br}^{\left(i\right)}-\rho_{br}^{\left(j\right)}\\ N_{br}^{\left(ij\right)}=N_{br}^{\left(i\right)}-N_{br}^{\left(j\right)}\\ M_{\varphi ,br}^{\left(ij\right)}=M_{\varphi ,br}^{\left(i\right)}-M_{\varphi ,br}^{\left(j\right)}\\ \varepsilon_{br}^{\left(ij\right)}=\varepsilon_{br}^{\left(i\right)}-\varepsilon_{br}^{\left(j\right)} \end{array}\right.$$
(3)$$\rho_{k}^{\left(i\right)}=\sqrt{{\left(x^{\left(i\right)}-x_{k}\right)}^{2}+{\left(y^{\left(i\right)}-y_{k}\right)}^{2}+{\left(z^{\left(i\right)}-z_{k}\right)}^{2}}$$
where λ is the wavelength of the carrier-phase signal; φ indicates the carrier-phase observation quantity; ρ is the distance of the satellite from the ground monitoring station; N is the ambiguity that needs to be calculated; M is the multipath error of the carrier phase observed by the receiver; ε denotes the receiver’s random noise error; $\left(x^{\left(i\right)},y^{\left(i\right)},z^{\left(i\right)}\right)$ are the coordinates of satellite i; $\left(x_{k},y_{k},z_{k}\right)$ are the coordinates of the monitoring station; and rb is the distance between the monitoring station and the reference station.

The carrier-phase double-difference observation model based on Equation (1) can be solved for a single epoch using the Extended Kalman Filter (EKF). Then, the integer ambiguity of the carrier phase is fixed using the Least-square Ambiguity Decorrelation Adjustment (LAMBDA) algorithm [10]. Finally, the remaining residual is the sum of multipath error and random noise [11]. Further processing with algorithms can weaken these errors and thus improve the accuracy of GNSS-RTK positioning.

### 2.2. The ICEEMDAN-AWPTD Combined Denoising Algorithm for Multipath Error Mitigation

#### 2.2.1. ICEEMDAN Algorithm

The ICEEMDAN algorithm provides a more accurate decomposition of signals into a sequence of IMFs arranged in descending frequency order. On the basis of the CEEMDAN algorithm, the ICEEMDAN algorithm further eliminates the residual noise in the IMF and solves the issue of false modes generated in the initial decomposition stage [12]. Figure 2 shows the ICEEMDAN algorithm flow chart.

In the ICEEMDAN algorithm, $E_{j}\left(\cdot \right)$ represents the jth IMF obtained by the Empirical Modal Decomposition (EMD) algorithm, $M\left(\cdot \right)$ is the operator that calculates the local mean of the signal, and $\varepsilon_{j}$ is the adaptively added white noise coefficient. In this case, $x\left(t\right)$ is defined as a signal sequence with processing, and white noise $v^{i}\left(t\right)$ is added to $x\left(t\right)$ to construct the following sequence:(4)$$x^{i}\left(t\right)=x\left(t\right)+\varepsilon_{0}E_{1}\left(v^{i}\left(t\right)\right)$$

The i local means in Equation (4) are calculated using EMD to obtain the first-order residuals ($r_{1}$):(5)$$r_{1}\left(t\right)=\langle M\left(x^{i}\left(t\right)\right)\rangle$$
where <·> is the operator that calculates the average value for the whole process.

By subtracting the original time series from $r_{1}$, the first intrinsic mode function $imf_{1}\left(t\right)$ of the signal to be decomposed is obtained:(6)$$imf_{1}\left(t\right)=x\left(t\right)-r_{1}\left(t\right)$$

For higher-order modes $\left(j=2,\;3,\ldots ,\;J\right)$, repeat the above steps and continue adding adaptive white noise to construct the sequence $r_{j-1}\left(t\right)+\varepsilon_{j-1}E_{j}\left(v^{i}\left(t\right)\right)$. Calculate its local mean to obtain the residuals of the jth stage ($r_{j}$), and then obtain the signal’s jth-order IMF and a residual ($R\left(t\right)$). At this point, the original time series ($x\left(t\right)$) can be expressed as
(7)$$x\left(t\right)=\sum_{j=1}^{J}imf_{j}\left(t\right)+R\left(t\right)$$

Regarding the two important parameter settings in the ICEEMDAN design, we adopted the experimental scheme in the literature [13], setting the mean number to 100 and the amplitude coefficient of the white noise to 0.2. By averaging 100 times, the influence of random factors can be reduced, and setting the white noise coefficient to 0.2 helps to better separate detailed information in the signal. The series of IMFs obtained from the decomposition reflect the information of the signal at different scales, so it is necessary to design a detailed classification method for the classification and processing of these IMFs.

#### 2.2.2. AWPTD Algorithm

The wavelet packet transform (WPT) is an extension of the discrete wavelet transform (DWT) that provides a more accurate local multi-resolution analysis capability for signal processing [14]. The wavelet packet threshold denoising (WPTD) algorithm uses a proper threshold function to quantize the wavelet coefficients in accordance with the characteristics of the wavelet coefficients and then reconstructs the processed wavelet packet coefficients to achieve overall signal noise reduction. In the WPTD algorithm, the choice of threshold function is critical and directly affects the final noise reduction effect of the algorithm. In this paper, we propose an adaptive wavelet thresholding function that is optimized in three aspects of the original thresholding function. The main points are as follows:(1)Ensure the continuity of the threshold function to avoid jumps and oscillations during the signal reconstruction;(2)Shrink the wavelet packet coefficients smaller than the threshold as much as possible while retaining the original information components as much as possible for coefficients larger than the threshold;(3)Adaptive adjustment according to the amount of noise in the wavelet packet coefficients.

The improved adaptive threshold function is as follows:(8)$$c\eta \left(w,t,s\right)=\left\{\begin{array}{cc} sign\left(w\right){\left({\left|w\right|}^{2}-t^{2}e^{\left(\frac{s-1}{s}\right)\left(\left|w\right|-t\right)}\right)}^{\frac{1}{2}} & \left|w\right|\geq t\\ 0 & \left|w\right|<t \end{array}\right.$$
where w is the wavelet packet coefficient before processing; η is the wavelet packet coefficient after processing; t is the threshold value; and s is the adjustable coefficient ($s\in \left(0,1\right]$). When the adjustable coefficient (s) approaches 1, the threshold function tends to be more biased towards soft thresholding, while it tends to be biased towards hard thresholding when the s value approaches 0.

In Figure 3, the threshold (t) is set to 0.3, and the improved threshold function is drawn as a graph.

If the wavelet coefficients contain a large amount of noise, adjust the s value to move it closer to the soft threshold function, which effectively compresses the noise signal. If the amount of noise is small, adjust the s value to move it closer to the hard threshold function, which better preserves the effective components of the original signal. The information entropy can reflect the amount of noise in a signal, so the coefficient (s) is adjusted to calculate the composite multi-scale permutation entropy (CMPE) [15].

The basic idea of the CMPE algorithm is to change the original time series $\left\{x_{i},i=1,\;2,\ldots ,N\right\}$ into a coarse-grained time series, and the formula is as follows:(9)$$y_{k,j}^{\left(\tau \right)}=\frac{1}{\tau }\sum_{i=\left(j-1\right)\tau +k}^{j\tau +k-1}x_{i},1\leq j\leq N/\tau ,1\leq k\leq \tau$$
where τ is the scale factor, which determines the length of each coarse-grained sequence ($y_{k,j}^{\left(\tau \right)}$).

Calculate the permutation entropy (PE) of the coarse-grained sequence $y_{k,j}^{\left(\tau \right)}$ under each scale factor (τ) and then average the PE values from the τ scales to obtain the CMPE values for each scale. The calculation formula is as follows:(10)$$CMPE\left(X,\tau ,m,\lambda \right)=\frac{1}{\tau }\sum_{k=1}^{\tau }PE\left(y_{k,j}^{\left(\tau \right)},m,\lambda \right)$$
where m is the dimensionality of the embedding and λ is the time delay.

By calculating the CMPE value of the wavelet packet coefficients, it is possible to obtain adjustable values of the parameter sequence $\left\{s_{i},i=1,2,3,\ldots \right\}$. These values can be plugged into Equation (8) to achieve an adaptive adjustment of the threshold function on the basis of the magnitude of the noise content of the signal.

#### 2.2.3. Multipath Error Extraction and Correction

Taking into account the periodicity of multipath errors in GNSS observations under static conditions, it is possible to extract and suppress these errors by continuously observing GNSS data for two consecutive days. To maximize the effectiveness of multipath error suppression, the systematic component of multipath error in GNSS data must be accurately extracted. This paper proposes a multi-scale anti-multipath denoising algorithm that uses modal function decomposition, effective coefficient sieving, and adaptive noise reduction to extract multipath error from GNSS observation data and then suppress multipath error to enhance data accuracy. The three-stage processing flow of the algorithm is shown in Figure 4.

The ICEEMDAN algorithm is capable of effectively analyzing nonlinear and non-stationary signals, enabling more accurate separation of the systematic and random components in GNSS observation data and providing a powerful tool for identifying the multipath error portion of the data. First, the ICEEMDAN algorithm is used to decompose the GNSS coordinate residuals ($Y\left(t\right)$) into a series of IMFs ranging from high to low frequencies:(11)$$Y\left(t\right)=\sum_{j=1}^{J}imf_{j}\left(t\right)+R\left(t\right)$$

The IMFs decomposed by ICEEMDAN contain all of the signal’s information, and a specific division criterion is required to determine which terms contain the multipath error information. In the second stage, a mode function division method based on information entropy is proposed to further partition modal components and connect the ICEEMDAN algorithm with the AWPTD algorithm. In Section 2.2, an information entropy method of CMPE was introduced to characterize the noise content of the signal. Similarly, the CMPE values are used here to set two metrics $(k_{1}$ and $k_{2})$ to separate the useful information part of the IMF from the random noise part. Based on the characteristics of information entropy, the more a time series exhibits randomness, the larger its entropy value for the random noise component. The CMPE values of several common signals are listed in Table 1.

IMFs with CMPE values above 0.65 have their wavelet coefficients almost completely compressed when wavelet packet threshold filtering is performed, so IMFs with CMPE values above 0.65 are used as the cut-off point to determine the first index value ($k_{1}$).

Among the decomposed IMFs, the CMPE values of the noise IMFs are relatively large and concentrated, while those of the useful signal IMFs are relatively small and evenly distributed. Based on these characteristics, Equation (12) is used to calculate the $E_{j}$ coefficients to determine the boundary point between the noisy IMF terms and the useful IMF terms.
(12)$$E_{j}=\frac{\left|cmpe_{j}-\frac{1}{j-1}\sum_{i=1}^{j-1}cmpe_{i}\right|}{cmpe_{j}},j\geq 2$$
where $cmpe_{j}$ is the CMPE value of the jth IMF. If all IMFs after the jth IMF are useful IMF terms, then the $E_{j}$ coefficient of this IMF should be greater than 1, whereas the $E_{j}$ coefficients of the previous j IMFs are all less than 1. The reason is that the first few IMF terms contain more noise and have similar CMPE values, whereas an $E_{j}$ value greater than 1 indicates that the CMPE value of the jth IMF is less than half of the average value of the CMPE of the first j−1 IMF terms, proving that the useful signal terms dominate from the jth IMF, and the index value $k_{2}$ is determined based on this rule.

The high-frequency to low-frequency IMF terms of the ICEEMDAN decomposition can be split into noisy terms ($Y{\left(t\right)}_{noise}$), mixed terms ($Y{\left(t\right)}_{mixed}$), and useful information terms $(Y{\left(t\right)}_{useful}$) by the two metrics ($k_{1}$ and $k_{2}$) defined in the second stage of the algorithm.
(13)$$\begin{array}{l} Y\left(t\right)=Y{\left(t\right)}_{nosie}+Y{\left(t\right)}_{mixed}+Y{\left(t\right)}_{useful}\\ \;=\sum_{j=1}^{k_{1}-1}imf_{j}\left(t\right)+\sum_{j=k_{1}}^{k_{2}-1}imf_{j}\left(t\right)+\sum_{j=k_{2}}^{J}imf_{j}\left(t\right)+R\left(t\right) \end{array}$$

The three terms of IMFs that result from the division of index values $k_{1}$ and $k_{2}$ are processed differently in the third stage. For the high-frequency noisy terms ($Y{\left(t\right)}_{noise}$), which contain almost no useful information components, it is possible to directly eliminate them; for the high-frequency mixed terms ($Y{\left(t\right)}_{mixed}$), some useful information remains, which cannot be directly eliminated but must be reduced to obtain the signal components; and for the useful terms ($Y{\left(t\right)}_{useful}$), it is possible to directly retain them for subsequent reconstruction without processing.

The AWPTD algorithm from Section 2.2.2 is employed here to reduce noise in the noisy terms $Y{\left(t\right)}_{mixed}$. It is processed using adaptive thresholding to reduce the noise content while keeping the original signal components to the greatest extent possible. Finally, the “clean” component of the denoised $Y{\left(t\right)}_{mixed}$ is reassembled with $Y{\left(t\right)}_{useful}$ to obtain the multipath error information ($\tilde{Y}\left(t\right)$) to be recovered.
(14)$$\tilde{Y}\left(t\right)=AWPTD\left(\sum_{j=k_{1}}^{k_{2}-1}imf_{j}\right)+\sum_{j=k_{2}}^{J}imf_{j}+R\left(t\right)$$

At the fixed monitoring stations, the environment at the measurement sites remains unchanged between adjacent days, while a satellite orbit has periodicity, resulting in a strong correlation of multipath error between adjacent days. Based on this, the proposed combined denoising algorithm can be used in three steps to weaken the multipath error in the observed data. First, extract the multipath error from the first-day coordinate residuals as the reference day signal ${\tilde{Y}}_{r}\left(t\right)$. Then, use the maximum correlation coefficient method to calculate the delay time of ${\tilde{Y}}_{r}\left(t\right)$ concerning $Y\left(t\right)$ of the second-day observation data. Finally, the sidereal filter is used to correct the next day’s coordinate sequence based on the amount of time elapsed and to reduce the effect of the multipath error.

## 3. Experimental Results

To validate the effectiveness of the proposed method in improving the multipath error of GNSS-RTK, both simulated data and real GNSS-RTK data were used to analyze the feasibility and effectiveness of the new method. The simulated data contained a variety of component signals and noise mixtures, which were used to analyze the denoising effect of the proposed method and validate its effectiveness in extracting the multipath error of GNSS-RTK. The real data were obtained by continuously observing data from a GNSS receiver for two consecutive days and were used to analyze the effectiveness of the proposed method in extracting and correcting multipath error in practical applications. In this experiment, the signal-to-noise ratio (SNR), root-mean-square error (RMSE), and Pearson’s correlation coefficient (R) were used to provide a more intuitive description of the effectiveness of the proposed method in reducing multipath error.

### 3.1. Data Simulation

The simulation data in the simulation experiment had a sampling interval of 1 s and consisted of a total of 5000 samples. The simulation data were composed of multiple harmonic signals, and different levels of Gaussian white noise were added during the simulation process to simulate different signal-to-noise ratios. Their expression was as follows:(15)$$\left\{\begin{array}{l} x\left(t\right)=s_{1}\left(t\right)+s_{2}\left(t\right)+s_{3}\left(t\right)+noise\\ s_{1}\left(t\right)=\sin\left(4\pi t/2000\right)\\ s_{2}\left(t\right)=\sin\left(3\pi t/800\right)\\ s_{3}\left(t\right)=\sin\left(5\pi t/600\right) \end{array}\right.$$
where $x\left(t\right)$ is the synthesized simulation data; $s_{1}\left(t\right)$, $s_{2}\left(t\right)$, and $s_{3}\left(t\right)$ are different frequency signals; and noise is Gaussian white noise. The simulation results are shown in Figure 5.

The following experimental methods were set up separately for comparison tests to better demonstrate the benefits of the proposed algorithm:(1)Use CEEMDAN algorithm;(2)Use WPTD algorithm;(3)Use ICEEMDAN-AWPTD algorithm.

In this experiment, the parameters of CEEMDAN in method 1 were the same as those of ICEEMDAN in Section 2.1. In method 2, the number of wavelet packet decomposition layers was set to five, the wavelet basis function was set to Sym6, the threshold rule was set to Heursure, and the threshold function was set to the hard threshold function.

First, three methods were employed to denoise the noisy signal, and the processed results are shown in Figure 6. In order to highlight the differences in the denoising results obtained with different algorithms, the residual results obtained by subtracting the denoised results of the three methods from the original signal are shown in Figure 7.

Table 2 presents the SNR, RMSE, and R values of the CEEMDAN, WPTD, and ICEEMDAN-AWPTD algorithms after denoising. As shown in Table 2, compared to the CEEMDAN algorithm, ICEEMDAN-AWPTD improved the RMSE by 22.8%, R by 2.8%, and SNR by 16.5%. Compared to the WPTD algorithm, ICEEMDAN-AWPTD improved the RMSE by 18.3%, R by 2.1%, and SNR by 12.4%. These results indicate that the proposed combined algorithm outperforms the single CEEMDAN and WPTD algorithms in denoising performance and has a good denoising effect. Therefore, it can be applied to extract multipath errors in GNSS observation data.

### 3.2. Analysis of Actual GNSS Measurement Data

The experiment used two low-cost u-blox M8T receivers. One was used as the base station, and the other was used as the monitoring station. The sampling epoch was set to 1 s, and the satellite elevation cut-off angle was set to 15°. The measured data were first processed using RTKLIB, and the coordinate residuals were obtained by taking the difference between the calculated coordinates and the known coordinates. It is generally believed that the coordinate residuals obtained using the carrier-phase double difference under an ultra-short baseline only consist of random noise and multipath error [16].

To compare the effects of multipath errors on GNSS-RTK observations in this experiment, the monitoring station was first placed in an open area for data collection with both stations fixed in place and a baseline length of approximately 7 m. The GNSS-RTK deployment is shown in Figure 8, and the observation results are shown in Figure 9.

In Figure 9, it can be observed that the horizontal and vertical accuracies of static GNSS-RTK observations in an open environment were approximately 5 mm and 10 mm, respectively, indicating high precision.

The satellite antenna is shown in Figure 10 below with a choke added, and the correlation between the residual coordinate data of the two adjacent days was relatively low.

The next experiment placed the monitoring station in an environment with obstacles, with the monitoring station approximately 2 m from the west wall and 3 m from the north wall, while the reference station remained in the same position. The baseline length remained at 7 m, and the deployment is shown in Figure 11.

The experiment was conducted during the same time period on two consecutive days: 27–28 October 2022 (DOY 270–271). The observation results are shown in Figure 12.

The results shown in Figure 11 indicate that when conducting GNSS-RTK observations in complex environments, the fluctuation of coordinate residuals increases and there is a certain correlation between the trends of coordinate residuals on adjacent days. In actual bridge monitoring, observation conditions are even more complex, and the low-frequency dynamic deformation and displacement of structures require high precision. However, due to the existence of multipath error, true low-frequency structural deformation data are obscured, making it difficult to meet the requirements of bridge health monitoring [17].

In this experiment, the GNSS-RTK monitoring station remained nearly stationary on two consecutive days: DOY 270 and DOY 271. As previously noted, multipath error exhibits a periodic repetition characteristic under static observation conditions. Therefore, the coordinate residuals between adjacent days have a strong correlation, and the time series of the first day relative to the second day is subject to a delay. Table 3 summarizes the maximum correlation coefficient of the coordinate residuals for DOY 270–DOY 271, along with the index value of the epoch where the maximum correlation coefficient occurred.

In Table 3, the correlation coefficients of the ENU coordinate residual series for adjacent days (DOY 270 and DOY 271) are all around 0.6, indicating a significant correlation between the coordinate series for the two days. Moreover, the DOY 270 coordinate series was delayed by about 4 min relative to the DOY 271 coordinate series, which was consistent with the theoretical time [18]. Therefore, the multipath errors in the DOY 270 coordinate residual can be extracted, and the extracted multipath error sequence can be used to correct the coordinate series observed on DOY 271, based on the delay amount at the epoch of maximum correlation. In short, the multipath errors in the reference day coordinate residual are first extracted, and then the subsequent day’s coordinate is corrected using the sidereal filtering method.

As a way of demonstrating the advantages of the proposed algorithm, we selected CEEMDAN, WPTD, and ICEEMDAN-AWPTD for comparison and analysis in the extraction of multipath errors from the DOY 270 coordinate residual sequence. In the comparative experiment, the parameter settings for CEEMDAN and WPTD were consistent with the simulation experiment in Section 3.1. The multipath error extracted using the three methods is shown in Figure 13.

Figure 11 indicates that the ICEEMDAN-AWPTD algorithm extracts multipath error with significantly less noise compared to the CEEMDAN and WPTD methods, and the extracted multipath error sequence does not jump, retaining more characteristic signal information. To provide a more intuitive analysis of the effectiveness of the three algorithms in extracting multipath error, Table 4 presents the coordinate residuals of the original DOY 270 data in the ENU direction and the RMSE values of the multipath error extracted using the three methods. Compared to the results, the RMSE values of the multipath error extracted using all three methods were smaller than those of the original observations, and the ICEEMDAN-AWPTD algorithm outperformed the CEEMDAN and WPTD methods.

This experiment also calculated the maximum correlation coefficient R values of DOY 270 and DOY 271 after the extraction of multipath error using CEEMDAN, WPTD, and ICEEMDAN-AWPTD, and the results are shown in Table 5. A larger R value indicates a higher correlation between the adjacent two-day coordinate sequences and more characteristic information extracted from the multipath error. The results in Table 5 show that the correlation coefficient of the multipath error model extracted using ICEEMDAN-AWPTD was slightly higher than those of the CEEMDAN and WPTD algorithms.

After conducting two comparative experiments, it can be demonstrated that the ICEEMDAN-AWPTD algorithm is effective in extracting multipath error from reference days while preserving the original information components in the signal as much as possible. Therefore, it is suitable for multipath error suppression.

The effectiveness of the ICEEMDAN-AWPTD algorithm for multipath error extraction from the coordinate residual sequence of the reference day (DOY 270) was demonstrated in the preceding comparison test, and subsequent experiments will validate the correction effect of the multipath error extracted by using this algorithm on the coordinate sequence of the subsequent second day (DOY 271). Similarly, we compared the performances of three methods (CEEMDAN, WPTD, and ICEEMDAN-AWPTD) in correcting the coordinate residual sequence of DOY 271 for multipath error, as shown in Figure 14.

The results in Figure 12 demonstrate that among the three methods compared, the ICEEMDAN-AWPTD algorithm was more effective in suppressing the amplitude of multipath error on the second day, and the remaining residual amount was almost random noise error.

To quantitatively analyze the correction accuracy of the three algorithms for multipath error, Table 6 presents the RMSE values of the coordinate residual sequences after processing for DOY 271. It is evident from Table 6 that the ICEEMDAN-AWPTD algorithm outperformed the CEEMDAN and WPTD algorithms for the multipath error reduction, with improvements of 49.2%, 65.1%, and 56.6% in the RMSE values of the original E, N, and U directions, respectively. The improvement was most noticeable in the N direction, which was due to the largest correlation coefficient of the adjacent two-day coordinate series in the N direction, indicating that it was more influenced by the multipath effect.

The results shown in Figure 12 and Table 6 indicate that the ICEEMDAN-AWPTD algorithm can extract multipath error with high accuracy and effectively mitigate the influence of multipath error in short-baseline GNSS-RTK observations.

## 4. Discussion

In Section 3, the feasibility of using the ICEEMDAN-AWPTD algorithm to reduce the impact of multipath error in GNSS-RTK monitoring was verified using simulation and actual measurement data.

In the simulation experiment in Section 3.1, the WPTD algorithm from the literature [5] and the CEEMDAN algorithm from the literature [6] were used for comparison. The experimental results showed that compared to the CEEMDAN algorithm, ICEEMDAN-AWPTD improved the RMSE by 22.8%, R by 2.8%, and SNR by 16.5%. Compared to the WPTD algorithm, ICEEMDAN-AWPTD improved the RMSE by 18.3%, R by 2.1%, and SNR by 12.4%. In the measured GNSS-RTK data in Section 3.2, the results showed that the coordinate accuracy in the E, N, and U directions improved by 42.9%, 54.5%, and 49.8%, respectively, after processing with the ICEEMDAN-AWPTD algorithm; by 45.8%, 58.9%, and 52.3%, respectively, after processing with the WPTD algorithm; and by 49.2%, 65.1%, and 56.6%, respectively, after processing with the CEEMDAN algorithm. Both the simulation and experimental results indicate that the ICEEMDAN-AWPTD algorithm outperforms the CEEMDAN and WPTD algorithms.

It was discovered through experimental comparison and analysis that while the wavelet packet hard threshold method in the literature [5] can better retain the original characteristics of the signal, the signal after noise reduction is prone to oscillation and contains residual noise. The CEEMDAN algorithm used in the literature [6] provides a method of IMF reconstruction based on effective coefficients, which divides the CEEMDAN decomposition’s IMF into noise and signal terms. However, the division threshold selection is primarily empirical, and some of the divided noisy terms also contain useful information, so their direct removal will affect the signal’s accuracy after noise reduction. The ICEEMDAN-AWPTD algorithm considers their benefits and improves their shortcomings. Firstly, this study introduced the ICEEMDAN signal decomposition algorithm to address issues with the CEEMDAN algorithm’s decomposition process. Secondly, we designed an IMF partitioning method based on information entropy, which directly and precisely utilizes the noise level of each IMF for partitioning. Finally, to preserve the original features of the signal to the maximum extent, we proposed an AWPTD algorithm to denoise the noisy components obtained from the decomposition, to some extent overcoming the limitations of the soft and hard threshold functions in denoising. Compared with the EEMD-WT method proposed in the literature [8], the ICEEMDAN-AWPTD combined denoising method exhibits superior performance in both modal decomposition and adaptive denoising.

## 5. Conclusions

To mitigate the effects of GNSS-RTK multipath error, this paper proposes a combined denoising algorithm using the ICEEMDAN-AWPTD algorithm to extract multipath errors and apply them for subsequent multipath error correction. This method adopts a three-level algorithmic processing approach to improve on the limitations of a single algorithm, ensuring the effectiveness and accuracy of the multi-scale separation of GNSS coordinate residual sequences to the greatest possible extent. The effectiveness of the ICEEMDAN-AWPTD combined denoising algorithm was verified through the analysis of both simulated and measured data. A comparative analysis showed that the method outperformed the CEEMDAN algorithm and the WPTD algorithm in terms of multipath error separation and correction, and improved GNSS-RTK positioning accuracy.

This method is mainly aimed at correcting multipath error in GNSS-RTK monitoring under static conditions, such as bridge deformation monitoring, where the target environment is almost constant, and has limited adaptability to monitoring under dynamic conditions. In addition, due to the special nature of the repetition period of the MEO satellite orbit in the BDS system, MEO-type satellites in the BDS system were not included in the observations. Its effectiveness in practical application is limited to a certain extent, and thus it needs further improvement in terms of theory and algorithm. In addition, due to the iterative operation required by the ICEEMDAN algorithm, which requires a long processing time and has poor real-time performance, it is more suitable for post-processing GNSS-RTK data.

## Figures and Tables

**Figure 1 sensors-23-08396-f001:**
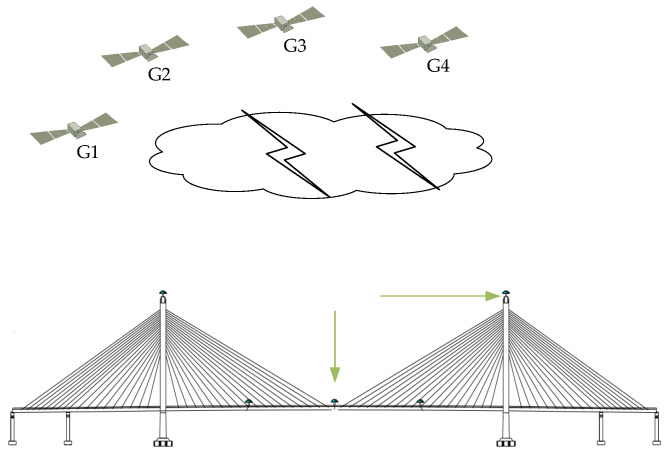
GNSS-RTK bridge monitoring schematic.

**Figure 2 sensors-23-08396-f002:**
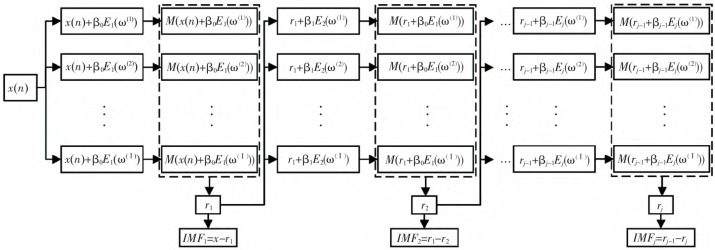
ICEEMDAN algorithm flow chart.

**Figure 3 sensors-23-08396-f003:**
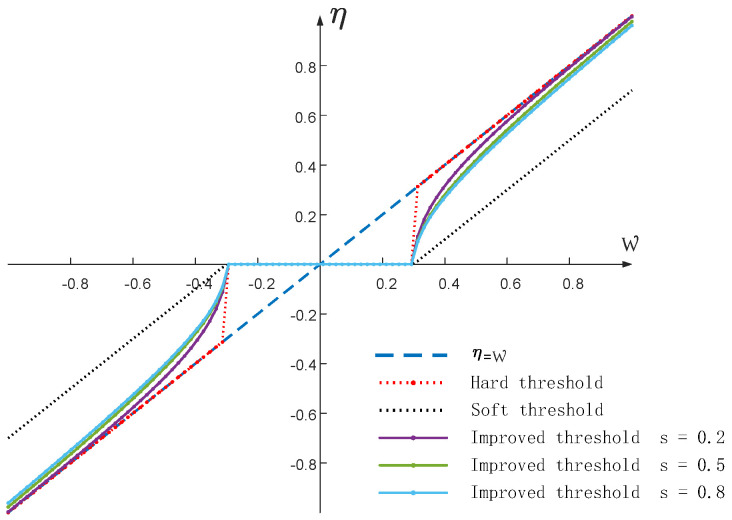
The improved adaptive threshold function.

**Figure 4 sensors-23-08396-f004:**
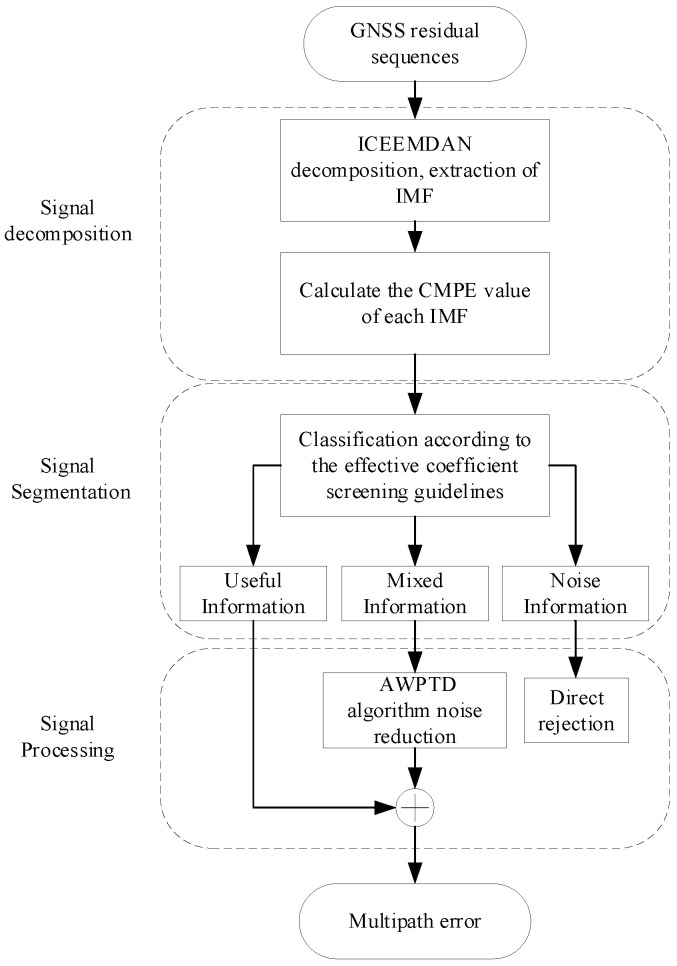
Multipath error extraction flow chart.

**Figure 5 sensors-23-08396-f005:**
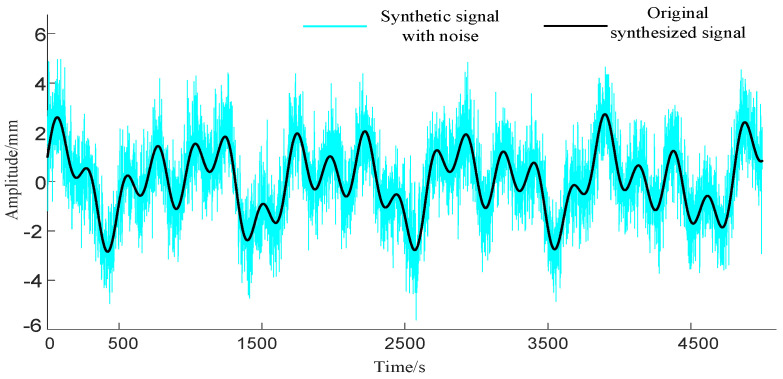
Raw signal and noisy signal.

**Figure 6 sensors-23-08396-f006:**
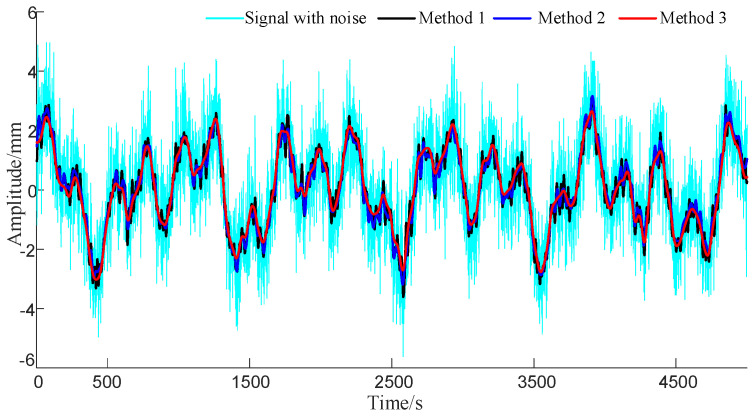
Comparison of noise reduction results of different solutions.

**Figure 7 sensors-23-08396-f007:**
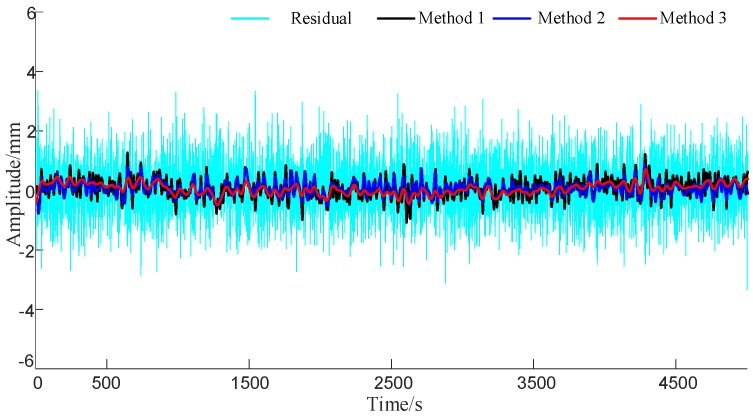
The residual values of denoising results from different methods and the original signal.

**Figure 8 sensors-23-08396-f008:**
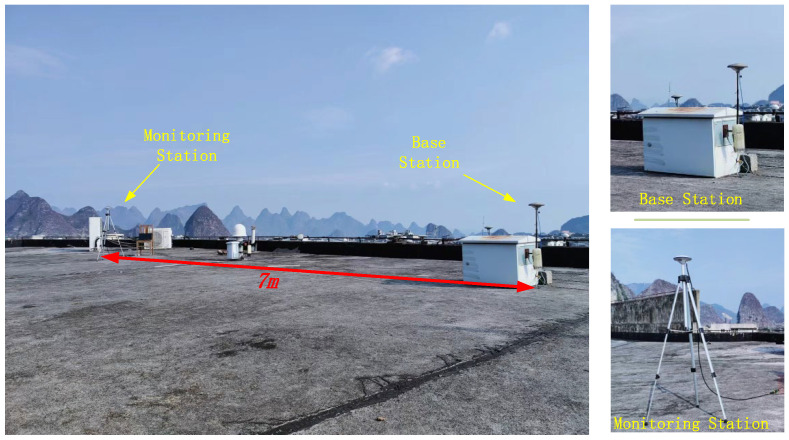
GNSS-RTK deployment map in an open environment.

**Figure 9 sensors-23-08396-f009:**
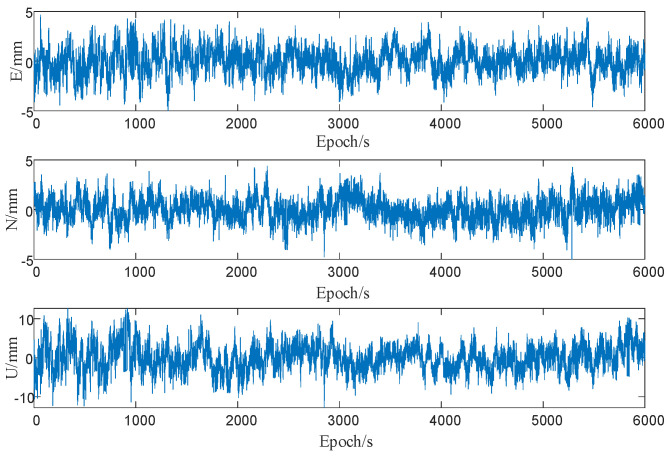
GNSS-RTK coordinate residuals in an open environment.

**Figure 10 sensors-23-08396-f010:**
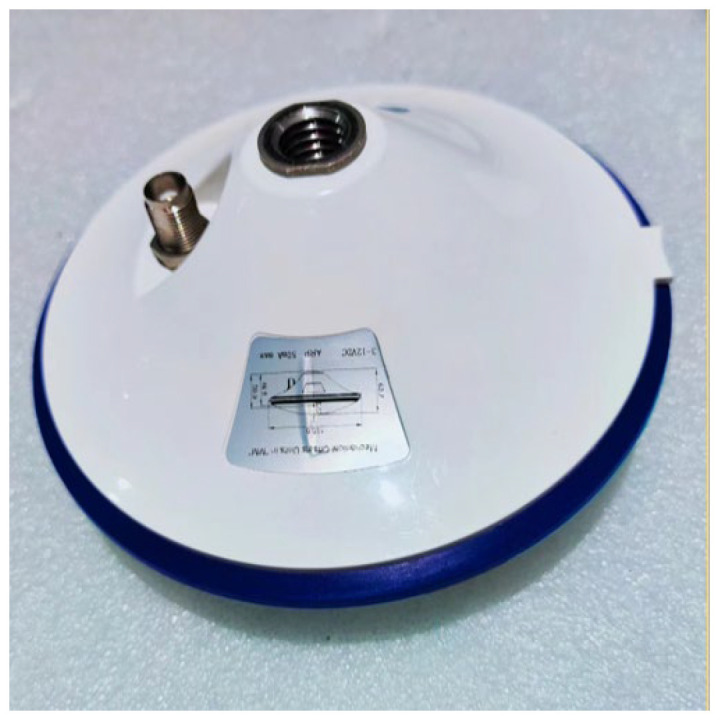
The satellite antenna.

**Figure 11 sensors-23-08396-f011:**
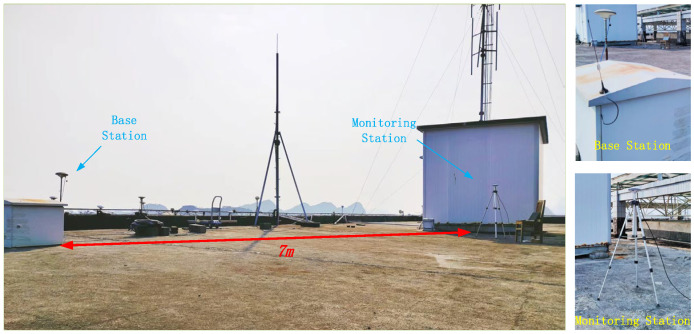
GNSS-RTK deployment map in the complex environment.

**Figure 12 sensors-23-08396-f012:**
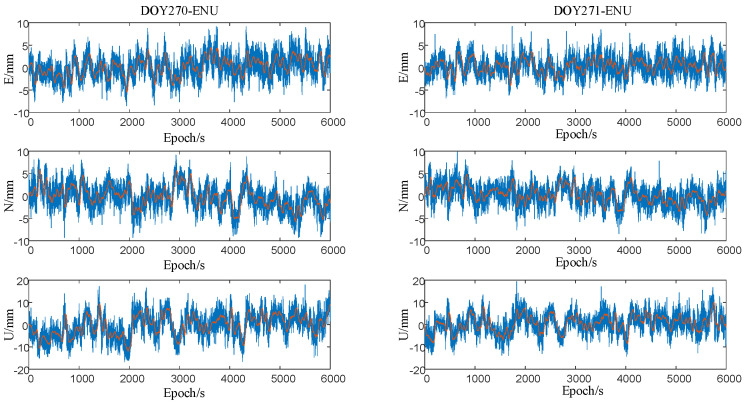
Adjacent two-day coordinate residuals (DOY 270–DOY 271).

**Figure 13 sensors-23-08396-f013:**
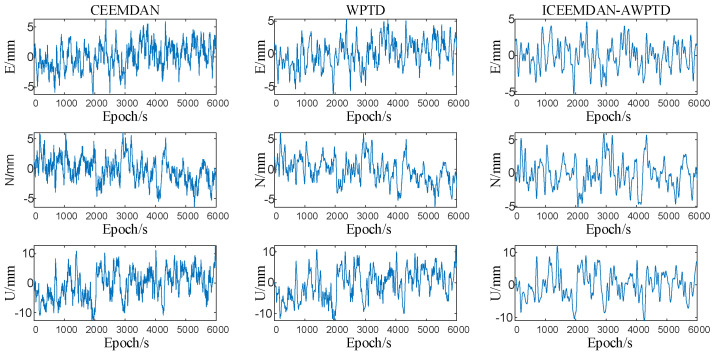
The multipath error values extracted using the three algorithms.

**Figure 14 sensors-23-08396-f014:**
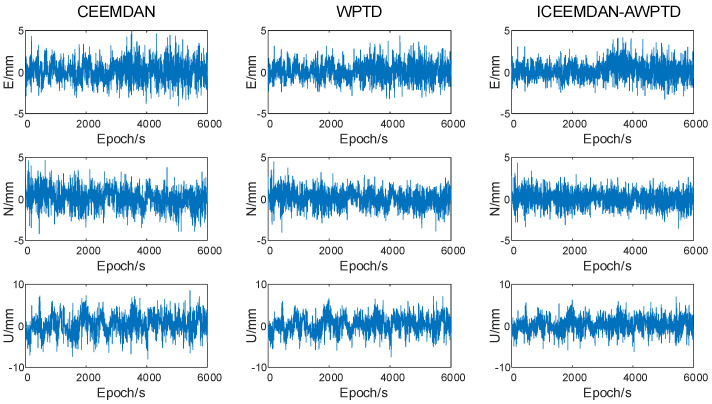
DOY 271 coordinate residuals after removing multipath error.

**Table 1 sensors-23-08396-t001:** CMPE values for several common signals.

Signal Type	CMPE Values
uniform white noise	0.964
Gaussian white noise	0.962
sine signal with noise	0.543
sine signal	0.241

**Table 2 sensors-23-08396-t002:** Statistics of noise reduction effects of three different solutions.

Method	RMSE (mm)	R	SNR
CEEMDAN	0.262	0.964	13.650
WPTD	0.248	0.972	14.144
ICEEMDAN-AWPTD	0.203	0.991	15.897

**Table 3 sensors-23-08396-t003:** Maximum correlation coefficient and corresponding ephemeral moments.

Dates	Coordinate	R	Epoch
DOY 270–DOY 271	E	0.615	−243
	N	0.638	−237
	U	0.596	−241

**Table 4 sensors-23-08396-t004:** RMSE values of extracted multipath error.

Method	RMSE (mm)		
	E	N	U
No filter	2.663	2.536	5.656
CEEMDAN	2.194	2.148	4.890
WPTD	2.094	2.063	4.417
ICEEMDAN-AWPTD	1.827	1.813	4.153

**Table 5 sensors-23-08396-t005:** Extraction of correlation coefficient values of two adjacent days after multipath model.

Method	R		
	E	N	U
CEEMDAN	0.718	0.752	0.654
WPTD	0.736	0.773	0.673
ICEEMDAN-AWPTD	0.768	0.807	0.694

**Table 6 sensors-23-08396-t006:** RMSE values and improvement effect of coordinate sequences before and after multipath error correction.

Coordinate (DOY 271)	RMSE (mm)				Improvement Effect (%)		
	No Filter	CEEMDAN	WPTD	ICEEMDAN-AWPTD	CEEMDAN	WPTD	ICEEMDAN-AWPTD
E	2.301	1.313	1.247	1.169	42.9%	45.8%	49.2%
N	2.432	1.109	0.997	0.849	54.5%	58.9%	65.1%
U	5.271	2.646	2.514	2.288	49.8%	52.3%	56.6%

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
