# Peer review of "A Multi-Scale Anti-Multipath Algorithm for GNSS-RTK Monitoring Application"

_sensors, 2023, doi:10.3390/s23208396_

Round 1

Reviewer 1 Report

The manuscript addresses a topic of great potential interest for GNSS surveying. This paper is a useful contribution and proposes an original approach to reduce the effects of multipath error in RTK-GNSS measurements.

The paper is mostly well written, but I have some comments, that are mostly editorial in nature:

Line 40: There is no space between the numerical value and the unit symbol.

Line 97: Replace "satellite. The" with "satellite, the".

Line 106: Replace "is" with "are".

Line 129: Replace "Where" with "where". Same in the lines: 163, 182, 188.

Line 138-140: It should be explained in more detail why these values were adopted.

Lines 328 and 340: It is claimed that the baseline length is approximately 7 m., but the FiG. 7 and Fig. 9 indicate that the length is 10 m.

All results reported should be based on the significant decimal places i.e. the reported accuracy should match the real accuracy of RTK measurements (Tables 2, 4, 6).

The main lack, in my opinion, is in the unclear replicability of the work. It is not obvious how to extend the method to reduce multipath errors in RTK-GNSS monitoring under dynamic conditions and in usual RTK surveys. It would be useful if the authors provide ideas on how to develop an efficient error correction model that can be applied in all situations.

Since I am not a native speaker, I cannot judge the language of the paper, but it is obvious that there is an unusual writing style in some cases, for example in lines 133-135, 173-176, 184-186, etc.

Reviewer 2 Report

  1. The proposed algorithm for reducing multipath error in RTK-GNSS measurements appears to be innovative and addresses an important issue in the field. The combination of improved complete ensemble empirical mode decomposition with adaptive noise (ICEEMDAN) and adaptive wavelet packet threshold denoising (AWPTD) seems promising.
  2. It is commendable that the algorithm incorporates modal function decomposition, coefficient sieving, and adaptive thresholding denoising to effectively mitigate the effects of multipath error. This multi-step approach shows a thorough consideration of the problem.
  3. The use of information entropy-based IMF selection is an interesting approach. It would be beneficial to provide more details on how this method accurately locates the IMFs containing multipath error information. Additionally, discussing the advantages and limitations of this selection method would enhance the clarity of the proposed algorithm.
  4. The optimization of the adaptive denoising method is crucial for preserving the original signal characteristics. It would be helpful to explain the specific optimization techniques employed and provide an evaluation of the results obtained. This will help readers understand the effectiveness and efficiency of the denoising process.
  5. The simulation data and RTK-GNSS measured data results indicate a higher accuracy of the multipath error correction model compared to singular filtering algorithms. However, more quantitative analysis and statistical metrics should be provided to support these claims. Additionally, discussing potential limitations and areas for further improvement would strengthen the research findings.
  6. It would be beneficial to include a discussion on the computational complexity of the proposed algorithm and any potential challenges in implementing it in real-time RTK-GNSS systems.
  7. Overall, the paper presents a valuable contribution to the field of RTK-GNSS measurements by addressing the issue of multipath error. With some minor revisions and additional clarity in certain sections, the paper has the potential to be a significant research contribution.
  1. Minor editing of the English language required

Reviewer 3 Report

The manuscript can be major revision for many reasons but these can generally be divided into technical reasons.

Introduction section needs revision. It should also introduce some latest research results in the domain, and motivation for the proposed work.

Literature review section must also be extended. Related works section may include literature survey. A comparative study may also be shown in graphical form.

Do you consider the topic original or relevant in the field? Does it address a specific gap in the field?

What specific improvements should the authors consider regarding the methodology? What further controls should be considered?

Add/cite recent publication (2020, 2021, 2022, 2023) preferably.

The overall structure of the article should be consistent: Abstract, Introduction, Materials and Methods (Here, include literature survey, background and proposed work), Experimental Analysis, Results, Conclusion, Future Work, References.

Language must be improved as there are linguistic errors at some places.               

Reviewer 4 Report

The study need to be significantly improved. Additional data 

acquisition needed and experiment design should be changed.

Round 2

Reviewer 3 Report

After reviewing all the suggestions made, I observe that they were fully carried out. Therefore, I have no more suggested changes.

Author Response

Thank you

Reviewer 4 Report

I could not find point-to-point response to my report for version 1 of the manuscript. Probably you missed, so I just put as is (note line numbering is as in version 1 of manuscript). Please provide point-to-point response: copy my comment and write you response below.

Round 3

Reviewer 4 Report

Thank you for providing point-to-point response. However I didn't notice any significant improvement to the text of manuscript. And general point of the response is not to improve manuscript according to the review but to make it publish in the form close to original.

Authors provide statement and does not provide any evidence. For example response to point 2:

Due to the repeatability of multipath errors, adjacent two day coordinate residuals will exhibit correlation. However, in normal positioning, regardless of the sampling results at any time, the correlation is extremely small and negligible, so there will be no situation where there is a high correlation coordinate residuals without multipath errors.

It could only be evident from the analysis of wide statistics which is not performed.

Another statements seem to be misleading in the context of current study entitled "Multi-Scale Anti-Multipath Algorithm for GNSS-RTK Monitoring Application". At the same time author state in the response that "We don't need to know how multipath errors move over time, which is not we need to focus on" (as a response to comment that multipath error should be detected using sliding window). From that I can assume authors don't have detection of the multi-path interval. Since correlation would be low for long time series where only small fraction have multi-path. So the algorithm authors introduce seems to work for data already selected to have multi-path.

Only single experiment is provided without flase-positive and false-negative control. Authors need significantly enhance experimental data volume and make description of methods and the result more clear.